# Biased, Bitopic, Opioid–Adrenergic Tethered Compounds May Improve Specificity, Lower Dosage and Enhance Agonist or Antagonist Function with Reduced Risk of Tolerance and Addiction

**DOI:** 10.3390/ph15020214

**Published:** 2022-02-10

**Authors:** Robert Root-Bernstein

**Affiliations:** Department of Physiology, Michigan State University, East Lansing, MI 48824, USA; rootbern@msu.edu

**Keywords:** bitopic, biased, cross-talk, tethered, enhancement, potentiation, specificity, tolerance, adrenergic, opioid

## Abstract

This paper proposes the design of combination opioid–adrenergic tethered compounds to enhance efficacy and specificity, lower dosage, increase duration of activity, decrease side effects, and reduce risk of developing tolerance and/or addiction. Combinations of adrenergic and opioid drugs are sometimes used to improve analgesia, decrease opioid doses required to achieve analgesia, and to prolong the duration of analgesia. Recent mechanistic research suggests that these enhanced functions result from an allosteric adrenergic binding site on opioid receptors and, conversely, an allosteric opioid binding site on adrenergic receptors. Dual occupancy of the receptors maintains the receptors in their high affinity, most active states; drops the concentration of ligand required for full activity; and prevents downregulation and internalization of the receptors, thus inhibiting tolerance to the drugs. Activation of both opioid and adrenergic receptors also enhances heterodimerization of the receptors, additionally improving each drug’s efficacy. Tethering adrenergic drugs to opioids could produce new drug candidates with highly desirable features. Constraints—such as the locations of the opioid binding sites on adrenergic receptors and adrenergic binding sites on opioid receptors, length of tethers that must govern the design of such novel compounds, and types of tethers—are described and examples of possible structures provided.

## 1. Introduction

This paper focuses on the observation that combinations of adrenergic and opioid drugs often display significantly enhanced activity that involves improved efficacy at lower drug dosages, increased duration of drug activity, and prevention of the development of tolerance. The mechanisms underlying this enhancement are reviewed and ways in which the enhancement mechanisms may provide the basis for the development of new classes of tethered bitopic drugs with biased activity and fewer side effects are addressed.

Opioids are analgesics often used to control pain perioperatively and postoperatively, as adjuncts to improve anesthesia during operations, and occasionally as stand-alone anesthetics [1,2]. They can also be used to treat refractory dyspnea [3,4]. Opioid antagonists are used to treat opioid overdoses and respiratory depression associated with opioid use or to reverse opioid analgesia [5]; antagonists are also used to treat opioid-induced constipation and post-operative ileus [6]. Some adrenergic agonists are also known to have analgesic activity [7,8,9] and can be used to treat pain disorders and opioid withdrawal symptoms [10,11]. However, adrenergic agonists are mainly used to treat a wide range of other conditions including asthma, chronic obstructive pulmonary disease (COPD), nasal congestion, bronchitis, emphysema, hypertension, hypotension, attention-deficit/hyperactivity disorder, shock, and cardiac arrest [12,13]. Adrenergic antagonists reduce the effectiveness of sympathetic nerve stimulation and are used to treat conditions such as angina, hypertension, arrhythmia, congestive heart failure, symptomatic benign prostatic hypertrophy, and adrenergic drug overdoses (e.g., over-medication for asthma) [12,13].

Various combinations of opioids and adrenergics demonstrate synergistic antinociceptive activity [14,15,16,17] that involves both α-adrenergic and β-adrenergic [18,19] components. For example, the addition of an adrenergic drug such as epinephrine or clonidine significantly reduces the amount of morphine required and its frequency of administration in the peri- and post-operative periods for a wide range of different types of surgeries [20,21,22,23,24,25,26,27,28,29]. The combination significantly decreases the need for intra-operative opiates [27,30], increases the duration of operative analgesia into the post-operative period [20,21,22,23,24,27,30,31]. These effects can be produced even when the delivery of the drugs is accomplished by separate routes at separate times [32,33]. Notably, several studies have demonstrated that the improved anesthesia and analgesia thus produced is not due to adrenergic-induced vasoconstriction in the operative area [29,34]. 

The α-2-adrenoceptor agonists epinephrine, clonidine, guanfacine, and dexmedetomidine inhibit the development of tolerance to morphine and other opioids [35,36,37,38]. One of the primary mechanisms of tolerance prevention appears to be opioid sparing, which has been observed with all four drugs [17,36,39,40]. Apparently, the necessity for less opioid in the presence of adrenergic agonists results in slower development of tolerance to the opioid [35,41,42,43], the mechanism of which will be discussed below. Notably, an adrenergic α(2C)-agonism/α(2A)-antagonism combination has proven to be particularly effective at preventing opioid dependence [44], suggesting that both agonists and antagonists may play important roles as enhancers of opioid therapy. An additional benefit of opioid–adrenergic agonist combinations is that the adrenergic agonist can reverse acute opioid tolerance, an effect established in both animal models and in the clinic [35,41,43,45,46,47,48]. 

Adrenergic antagonists can also potentiate opioid activity [49]: opioid sparing and decreased tolerance to the opioid has been reported in the presence of prazosin (an α adrenergic antagonist) [50]; ultra-low doses of the non-specific α adrenergic antagonist BRL44408 [51]; phentolamine, an α-blocker, or propranolol, a β-blocker [52]; and the β1-specific adrenoceptor blocker, esmolol [53,54,55,56,57]. Idazoxan and other I2-imidazoline ligands that are highly selective α2-adrenoceptor antagonists also attenuated morphine tolerance in rats [58]. Many of these studies report opioid sparing and resistance to the development of opioid tolerance as beneficial effects of the presence of the adrenergic antagonists. This enhancement by adrenergic antagonists strongly argues that an adrenergic receptor-mediated mechanism of action for this enhancement is unlikely.

In sum, α- and β-adrenergic agonists and antagonists can combine with opioid drugs to produce enhanced analgesia and to maintain it, sparing further opioid use, and together prevent the development of opioid tolerance (summarized in Table 1). Despite these benefits, Chabot-Doréet al. [59] report that combinations of adrenergic and opioid drugs are “sadly underutilized” in anesthesia, an observation validated more recently by Thiruvenkatarajanet al. [60]. One reason may be ignorance: the literature on adrenergic-opioid enhancement effects is widely scattered and tends to focus on very specific uses of the combinations for particular types of surgery. Another possible reason is that adrenergic enhancement of opioids does not extend to all combinations so that experimentation is required before clinical application becomes useful. For example, Fournier et al. [34] found that neither epinephrine nor clonidine potentiated infusions of the opioid sufentanil following hip surgery. The α adrenergic agonist dexmedetomidine enhanced morphine antinociception in spinal cord through mu opioid receptors but did not enhance DSTBULET (Tyr-D-Ser(OtBu)-Gly-Phe-Leu-Thr) activation of delta opioid receptors [61] while guanfacine (another α adrenergic agonist) potentiated deltorphin opioid activity at delta-opioid receptors (DOR) but not morphine or DAMGO ([D-Ala2, N-MePhe4, Gly-ol]-enkephalin; a synthetic opioid peptide) opioid activity via mu opioid receptors (MOR) [62]. DOR ligands, but no mu or kappa opioid ligands, specifically enhanced β adrenergic receptor function in intrinsic cardiac adrenergic cells to augment cardiac contraction [63]. Furthermore, adrenergic potentiation of opioids does not extend to all physiological systems, which can be seen as both a possible problem and a possible benefit. For example, Hughes et al. [64] found that no class of adrenergic drug affected morphine-induced changes in discriminative behaviors of rats, while Zarrindast et al. [65] reported that activation of α2-adrenergic pathways contributes to morphine-induced Straub tail (a condition in which an animal carries its tail erect), while α1- and β2-adrenergic pathways did not. In other words, specific pairs of adrenergic and opioid drugs need to be employed for particular purposes and targeted at particular pairs of adrenergic and opioid receptors in different tissues or organ systems. An additional potential concern that may prevent some practitioners from utilizing opioid–adrenergic combinations is that adrenergic drugs can have adverse systemic effects of their own, such as increased heart rate and blood pressure, that may counter-balance the analgesia-enhancing benefits in particular patient populations [7,8,9,17,39]. Notably, however, some studies have found that combining adrenergic drugs with opioids actually decreases the risks of cardiovascular effects [53] (Yu et al., 2011). Finally, an additional risk of combining opioids with adrenergic drugs is the cross-tolerance that develops between the drugs, so that over-use of adrenergic agonists can itself lead to withdrawal symptoms [66,67,68]. Despite these potential drawbacks, the literature on opioid–adrenergic combinations is broadly positive about their benefits. 

The thesis of this paper is that tethering opioid drugs to adrenergic drugs may permit novel bitopic, biased compounds to be designed that have improved specificity for particular receptor sets decreasing the dosage required to achieve pain management through enhanced activity and increased duration of activity. Lower drug doses should result in decreased side effects and, most importantly, less risk of resulting in development of tolerance and/or addiction to either the opioid, the adrenergic drug or the combination. Biased binding of such tethered drugs should also decrease side effects and permit better control of the resulting analgesia. The specific mechanisms underlying such mutual potentiation of activity and the criteria for the design and testing of such bitopic, biased, tethered compounds are explored below and the ways in which these mechanisms can be used to design novel drugs described. 

## 2. Mechanisms Underlying Opioid–Adrenergic Enhancement

Both opioid and adrenergic receptors belong to the broad class of G-protein coupled receptors (GPCR) that are activated by binding an appropriate ligand into their highly specific orthosteric binding site. Orthosteric ligand binding results in a conformational change in the receptor that allows an intracellular guanine nucleotide-binding protein (G protein) to exchange a guanine diphosphate (GDP) for a guanine triphosphate (GTP), the GTP then permitting one of the G protein subunits to dissociate and act as an intracellular second messenger to activate downstream functions. The dissociation of the G protein also activates the phosphorylation of the remaining subunits by G protein-coupled receptor kinases (GRKs), which results in the internalization of the receptor, and thus the downregulation of receptor activity. Repeated or continuous stimulation of GPCR results in tolerance to the ligand as a result of this downregulation so that increasing concentrations of drug are needed to produce the previous effect [69]. 

GPCR are also well-known to have both extracellular and intracellular allosteric binding sites that can enhance receptor function by locking the receptor into its high affinity state in the presence of its orthosteric ligand or that can impair receptor function by interfering with ligand binding or activation [70]. As noted above with regard to opioid–adrenergic drug combinations, the result of this allostery is to produce increased orthosteric ligand activity (whether agonistic or antagonistic) and to decrease downregulation of the receptor thereby extending the duration of activity. Thus, GPCR allosteric mechanisms have been the targets of drug development for many years [71,72,73,74,75,76]. However, surprisingly little research has been done on allosteric drugs for opioid receptors and even less on the role of adrenergic drugs as allosteric modifiers of opioid receptor function, the assumption apparently being that their interactions can be explained by intracellular “crosstalk” through heterdimerization of the receptors or at the second-messenger level [77,78,79]. However, recent evidence suggests that one mechanism underlying the mutual enhancement of opioid and adrenergic drugs for each other is binding of each drug to such an extracellular allosteric site on the other’s receptor. A second enhancement mechanism involves potentiation of hetero-oligomerization of the receptors.

Evidence of opioid binding to adrenergic receptors and of adrenergic drugs to opioid receptors comes from diverse types of experiments. Opioid drugs have been demonstrated to bind to adrenergic receptors. Indirect evidence of such binding comes from studies showing that morphine, naltrexone, and naloxone inhibited the binding of [3H]clonidine and [3H]-epinephrine to α 2-adrenoceptors in their high-affinity state [80]. Similarly, the MOR agonist tramadol has been found to have non-opioid receptor activity that can be blocked by the α-adrenergic antagonist yohimbine [81]. Subsequent studies have demonstrated that morphine and related opioids such as meperidine, remifentanil, and tramadol (but not sufentanil) bind directly to α2-adreneceptors displacing adrenergic agonists and generally exhibiting a higher affinity for the α(2B) and α(2C) than for the α(2A) subtype [82,83]. In contrast, fentanyl and naloxone did not bind to α(2)-adrenoceptors or displace their ligands [82,83]. Opioid agonists, including morphine and met-enkephalin (MENK) and antagonists such as naloxone also bind to β adrenergic receptors, the binding site having been identified as including the second and third extracellular loops of the receptor [84,85,86]. The studies cited in this paragraph demonstrate that the result of cooperative binding opioids with adrenergic enhancers is to decrease the concentration of opioid required to produce receptor activation and to increase the duration of that activity.

Similarly, epinephrine binds directly to the MOR enhancing morphine binding [85,86]. Other adrenergic compounds that bind to MOR with high affinity include norepinephrine and amphetamine [85,86]. Dopamine, salbutamol, and propranolol were found to have moderate affinity for MOR, while yohimbine and phentolamine had insignificant binding [85,86,87]. The binding sites for adrenergic compounds on the MOR are once again located in the second and third extracellular loops of the receptor [84,85,86] and produced enhanced activity at lower adrenergic concentrations along with increased duration of activity [88]. 

Evidence from binding studies demonstrates that the extracellular binding sites on both the adrenergic and opioid receptors share, to some extent, affinity for both opioids and adrenergic drugs [85,86,87,89]. This observation is consistent with the observation that there are metastable binding sites for opioids [90,91,92,93] and for adrenergic drugs [74,94,95] located in the extracellular regions of both types of receptor that may act as a sort of staging platform for guiding the ligand into its high-affinity orthosteric pocket. Indeed, the mutual affinity of opioid receptors for adrenergic compounds and adrenergic receptors for opioid compounds appears to be due to the origins of both receptor classes from a common evolutionary origin [89]. 

One of the most important aspects of mutual opioid–adrenergic allosteric enhancement of each other’s receptors is that this enhancement can be performed by antagonists (e.g., naloxone) as well as agonists (e.g., morphine), which argues against the enhancement resulting solely or even mainly from second-messenger crosstalk between the receptors or activation of both opioid and adrenergic receptors concurrently (though both mechanisms may nonetheless contribute to enhancement effects). Reference to the previous section demonstrates that such antagonist-induced enhancement has been documented in some clinical studies and it is worth emphasizing here that it has been very well established in experimental settings. For example, a series of studies have demonstrated that naloxone, an opioid antagonist, can enhance epinephrine and isoproterenol activity on both cardiac and smooth muscle preparations by mechanisms that do not involve the opioid receptor [96,97,98,99,100,101,102]. Similar data demonstrating that β-adrenergic antagonists [52,53,54,55,56,57,103] and α-adrenergic antagonists [50,51,52,58,103] enhance opioid efficacy confirm that the opioid sparing effects and inhibition of tolerance caused by adrenergic drugs and their increase of the duration of opioid analgesia must also be, at least in part, due to direct effects on the opioid receptor rather than through second-messenger mechanisms of cross-talk with adrenergic receptors. 

One of the explicit assumptions of this review is therefore that allosteric binding sites for adrenergic compounds exist on the extracellular side of the orthosteric opioid binding site (Figure 1, Figure 2 and Figure 3) as imputed from the binding studies and enhancement documented above. Notably, Traynor’s group [104,105,106,107], Filizola’s group [108,109,110], and Matosiuk’s group [111,112] have each studied small molecule, non-adrenergic allosteric enhancers of opioid receptors and, using molecular dynamics studies, also located allosteric binding to a site extracellular to the orthosteric site and often involving transmembrane-to-extracellular loop regions as illustrated in Figure 1, Figure 2 and Figure 3. Additionally, Uprety et al. [113] demonstrated the existence of a computationally, synthetically, and pharmacologically validated allosteric binding site in MOR and KOR that involves the same transmembrane 5-extracellular loop 2 region identified in adrenergic binding studies (above). Thus, binding, synthesis, and modeling studies agree. That said, it is important to stress that the exact location of the allosteric binding site may differ from one ligand to another and has not, in any instance, been unequivocally determined using methods such as cryogenic electron microscopy, X-ray crystallography, or cross-linking and thus remains a testable prediction of this review.

Having established that adrenergic-opioid enhancement can be mediated by allosteric mechanisms activated by antagonists as well as agonists, it is important to stress that hetero-oligomerization of opioid and adrenergic receptors has been well-established and also plays an important role in mediating cross-talk between these receptors (Figure 4) [59,115,116,117]. Thus, while second-messenger crosstalk mechanisms are not required to produce opioid and/or adrenergic receptor enhancement, such second-messenger crosstalk contributes to such enhancement effects. Evidence for such hetero-oligomerization-specific crosstalk is legion. For example, morphine and tramadol analgesia by clonidine can be blocked by the α-adrenergic antagonist yohimbine [118,119]. Yohimbine, notably, has been found to have no significant affinity for opioid receptors [94,95,96] so that its antagonism must be produced via its effects on the adrenergic receptor, which in turn argues for an adrenergic-receptor-mediated enhancement of opioid receptor function. Similarly, the β2 adrenergic agonists terbutaline and formoterol have been found to relieve neuropathic pain in mice but this effect disappeared in δ-opioid (but not μ-opioid and in κ-opioid) receptor deficient mice [120]. Furthermore, β2 adrenergic-induced pain relief could be blocked specifically by the δ-opioid receptor antagonist naltrindole as well as by the non-specific opioid antagonist naloxone. Thus, the effects of β2-agonists in a context of neuropathic pain can be argued to require peripheral δ-opioid receptors [120]. Examples such as these clearly make a case for opioid-receptor–adrenergic-receptor hetero-oligomerization as a second mechanism by which opioid–adrenergic co-potentiation can be achieved. 

Some studies (e.g., [19]) suggest that both allostery and hetero-oligomerization mechanisms work hand-in-hand and that, whichever mechanism is involved, the synergy that results is mediated by these mechanisms’ effects on G protein function. Vulliemozet al. [121] demonstrated that α-2 adrenergic- and opioid-receptors are coupled to each other through the cGMP effector system. Sullivanet al. [61] found that the synergy between MOR and α-adrenergic receptors was mediated by inhibition of the protein kinases that phosphorylate G proteins and Roeriget al. [122] found a similar mechanism behind delta opioid synergy with α-adrenergic receptors. Parket al. [123] also found that naloxone’s enhancement of β-adrenergic activity was mediated by its antagonism of G protein phosphorylation by GRK. Thus, modifications of receptor conformation, whether caused by the binding of allosteric enhancers, promotion of hetero-oligomerization of the receptors, or both, results in inhibition of G protein-mediated downregulation of the receptors and the consequent functional enhancement of their orthosteric ligands.

## 3. Designing Heterobitopic Agonists, Antagonists, or Mixed Action Drugs to Optimize Opioid–Adrenergic Enhancement

Bitopic compounds that increase GPCR receptor selectivity by linking a compound that binds to the orthosteric site with another that binds to the allosteric site have previously been designed and tested successfully (reviewed in [74,76,124]). In particular, homobitopic compounds (composed either of agonists or antagonists) that take advantage of the metastable binding site have successfully been designed for adrenergic receptors [94,125,126] and, separately, for opioid receptors [127,128,129,130]. Additionally, some bitopic opioid compounds with improved specificity for mixed-action KOR/MOR have been produced [131] and for MOR [132,133] as well as bitopic adrenergic compounds with similarly improved specificity [71,74,94,134]. However, it appears that heterobitopic ligands involving combinations of opioids and adrenergics have not yet been synthesized or tested. Thus, the mutual opioid–adrenergic enhancement mechanisms described above have not yet been exploited for their full pharmaceutical potential and therein lies the novel potential of bitopic, tethered opioid–adrenergic compounds.

Since at least two mechanisms exist by which opioids and adrenergic drugs enhance each other, one being through an allosteric mechanism and the other through hetero-oligomerization, it follows that there are at least two different ways of designing heterobitopic compounds to exploit these mechanisms. Consider the allosteric mechanism first.

Tethered opioid–adrenergic bitopic allosteric modulators will take advantage of the proximity of the orthosteric and allosteric binding sites that are present in both the opioid and adrenergic receptors (Figure 5). Because these binding sites are close together, tethers may be only a few atoms in length, as is illustrated by 17-cyclopropylmethyl-3,14β-dihydroxy-4,5α-epoxy-6β- [(2′-indolyl)acetamido]morphinan (INTA), a morphine agonist potentiated by conjugation to an indole moiety that showed potent antinociception and had no tolerance, dependence, and aversive effects in the conditioned place preference assay [132,133]. Evidence from homobitopic opioid and adrenergic compounds also suggests that tethers are likely to be short, consisting of only a few atoms in length [74,132,133,134,135]. Examples of such tethers include polyethylene glycol (PEG) [114,136,137], or short peptides such as polyproline [138] and polyglycine [139] or their modifications [139], usually two or three amino acids in length. Fultonet al. [139] demonstrated that adding hydroxyl groups to hydrocarbon-based linkers significantly improved binding of divalent, tethered compounds while Deckeret al. [128] found that adding methyl groups adjacent to the hydrolytically labile ester linkage increased stability. Additionally, calculations by Bobrovnik [140] strongly suggest that flexible linkers are likely to produce far greater increases in bivalent ligand binding at low concentrations of the tethered drug than can be achieved either by rigid linkers or by exposure of the receptor to each the ligand and enhancer separately. Obviously, an almost unlimited number of permutations of simple, flexible polymers with hydroxyl and methyl groups in appropriate places that are tailored to, and optimized for, particular GPCR are possible and it is clear from existing examples that some of these produce better receptor activation and stability than others [74,114,124,132,133,134,135,136,137,138,139,140]. Figure 6 and Figure 7 provide examples of possible heterobitopic opioid–adrenergic agonist compounds utilizing morphine or methionine-enkephalin as the opioid agonists and epinephrine as the adrenergic agonist. While peptide opioids, such as the enkephalins, may not themselves be useful as therapeutics, they have been used successfully as components of bitopic compounds [107,141] for investigative purposes, and more stable versions, such as D-Ala2, N-MePhe4, Gly-ol]-enkephalin (DAMGO), may provide better medicinal chemistry options.

Heterobitopic tethered inhibitors or antagonists for opioid and adrenergic receptors are also possible. Indeed, the bitopic approach to drug design originated in Portoghese’s synthesis of bivalent compounds with improved naltrexamine antagonism for opioid receptors [73,142], which became the model for the development of many subsequent bitopic opioid antagonists with improved specificity and efficacy [124]. Improved adrenergic antagonists have also been designed. For example, a bitopic compound consisting of a pair of yohimbine analogs tethered by three methylenes or methylene-diglycine acted as a highly selective human α2C-adrenergic receptor ligand [126,127]. Thus, it is reasonable to propose that tethered compounds consisting of an opioid antagonist such as naloxone or naltrexone tethered to an adrenergic antagonist such as yohimbine would result in enhanced inhibition of receptor activity and also permit antagonism of specific adrenergic, and possibly opioid receptor subtypes, as is illustrated in Figure 8: The untethered mixture of yohimbine and naloxone has previously been shown to have such enhanced antagonistic activity by Vo and Drummond [143,144]. 

Additionally, mixed function heterobitopic compounds may be designed that potentially have a series of novel functional possibilities. For example, since opioid antagonists enhance adrenergic receptor activity (see above), tethering an opioid antagonist to an adrenergic agonist should result in enhanced antagonism of the opioid receptor but enhanced adrenergic receptor activity. Such a drug might be useful, for example, in the treatment of hypotension, negative inotropic and chronotropic effects, as well as a decrease in baroreceptor reflex responses due to an opioid overdose. Alternatively, since opioid agonists enhance adrenergic antagonist activity (see above), tethering an opioid agonist to an adrenergic antagonist may result in an opioid with enhanced agonist activity as well as enhanced adrenergic antagonist activity. Such a drug might be useful in maximizing opioid-induced analgesia while minimizing adrenergic side-effects, such as increases in heart rate and blood pressure. 

It must be emphasized that one benefit that can be expected for all possible tethered combinations of agonist and antagonists just described is a significant decrease in the concentration of drug required to achieve a clinical outcome along with greater specificity of action and therefore a decrease in unwanted side-effects [107]. As Vauquelin [145] comments

“Bivalent ligands often display high affinity/avidity for and long residence time at their target. The [mechanism] responsible is the synergy that emanates from the simultaneous binding of their two pharmacophores to their respective target sites…. The first binding event prompts the still free pharmacophore to stay into ‘forced proximity’ of its target site, such lanes can be looked into by the equations that also apply to induced fit binding mechanisms. Interestingly, the simplest equations apply when bivalency goes along with a large gain in avidity. The overall bivalent ligand association and dissociation will be swifter than via each lane apart, but it is the lane that allows the fastest bidirectional ‘transit’ between the free and the fully bound target that is chiefly solicited. The bivalent ligand’s residence time is governed not only by the stability of the fully bound complex but also by the ability of freshly dissociated pharmacophores to successfully rebind.”

He goes on to point out that thermodynamic “simulations reveal that positive cooperativity exacerbates these phenomena, whereas negative cooperativity curtails them,” so that both agonistic and antagonistic activity are significantly increased [145] (see also [146]). 

Notably, the increases in agonist or antagonist activity just described can be expected whether the bivalent compound works within a receptor at orthosteric and enhancement binding sites, across orthosteric sites on dimerized receptors, or through some combination of both (Figure 9). While the vast majority of GPCR bitopic compounds that have been synthesized and tested thus far appear to exploit the allosteric mechanism just described, a few investigators have explored the potential of bitopic compounds to enhance or stabilize receptor oligomerization instead. For example, Lalchandaniet al. [125] found that a tethered bitopic compound consisting of a pair of yohimbine analogs tethered by 24 methylene spacers acted as a highly selective human α2C-adrenergic receptor ligand that increased receptor activity (as measured by cAMP production) more than 50-fold by promoting homodimerization of the receptor. Thus, it should be possible to produce heterobitopic opioid–adrenergic drugs with very long tethers designed specifically to enhance heterodimerization of specific opioid and adrenergic receptors (Figure 9 and Figure 10).

Mixed agonist–antagonist bitopic compounds have also been explored and provide models for opioid agonist–adrenergic antagonist or opioid antagonist–adrenergic agonist tethered compounds [147,148,149,150]. For example, the κ-selective antagonist pharmacophore 5-guanidinonaltrindole tethered to the δ-selective antagonist pharmacophore naltrindole spans the distance separating the orthosteric bindings sites of the opioid receptor heterodimer using linkers that were 20 to 21 atoms in length [151,152]. The resulting compound, KDN-21, stabilized the heterodimer resulting in greatly enhanced function [151,152]. 

Another example of a dimerization-enhancing bitopic, tethered drug is provided by Peng et al. [127], who designed bivalent ligands containing butorphan linked to nalbuphine, naltrexone, or naloxone with varying degrees of specificity and activity at MOR, DOR and KOR. One of these compounds, for example, exhibited κ opioid receptor agonist/antagonist and μ opioid receptor antagonist activity, while another was a κ agonist and μ agonist/antagonist and both modulated heterodimerization of opioid receptors (Figure 11). Thus, such combinations of agonists and antagonists in the same molecule may provide ways to target specific types of opioid–adrenergic receptor hetero-oligomerization and consequent functionality. Such biased opioids may provide targeted and analgesia and antinociception at significantly lower drug doses that have less probability of developing tolerance and are therefore much safer than non-specific opioids.

## 4. Conclusions

To summarize, extensive experimental and clinical data demonstrate that combinations of opioids and adrenergic drugs often result in enhance opioid activity characterized by a significant decrease in the amount of opioid required to attain analgesia and antinociception; such combinations increase the duration of opioid activity, inhibit development of tolerance and, in some cases, have been demonstrated to reverse acute tolerance. One drawback of these combinations is that they often have broad systemic effects and can activate multiple receptor subtypes. The present paper suggests that better specificity, increased activity, and decreased side-effects may be achieved by tethering the adrenergic and opioid components of such mixtures. Surprisingly, no examples of such tethered adrenergic-opioid compounds have been found in the literature. Moreover, while many adrenergic-opioid drug mixtures have been explored that may provide the basis for deriving such novel tethered compounds, many possible combinations of opioids and adrenergic drugs have not yet been explored, and certainly not in any systematic way, leaving a very large set of possible new drug entities.

Table 2 summarizes the possible options for biasing tethered opioid–adrenergic drugs. Combining an opioid agonist with an adrenergic agonist should (according to existing mixed treatment studies) yield tethered drugs that have enhanced agonist activity at both opioid and adrenergic receptors, further augmented by stimulation of both homodimerization and heterodimerization of the receptors. Such a combination might be extremely useful for localized analgesia and antinociception such as in spinal anesthesia or topical anesthesia in the context of asthma, COPD, shock, or cardiac arrest. Combining an opioid antagonist with an adrenergic agonist should yield enhance antagonism at both receptor types along with interference with both homo- and heterodimerization. Such an enhanced antagonist could be extremely useful for treating opioid overdoses or the side effects of prolonged opioid use where it would be deleterious to activate adrenergic functions or where an enhanced adrenergic antagonist would be a benefit (e.g., in treating an individual with angina, heart failure or benign prostate hyperplasia). It is also possible produce tethered compounds that enhance the opioid activity without activating adrenergic receptors (opioid agonist–adrenergic antagonist compounds) and, similarly, ones that enhance the adrenergic activity without activating opioid receptors (opioid antagonist–adrenergic agonist compounds). Data on how these mixtures would affect heterodimerization of receptors is generally lacking so that the table provides predictions that are logical but may, or may not, be accurate in practice. The utility of such mixed enhancer-antagonist tethered compounds might be found in targeting the drugs toward particular cellular or tissue targets while limiting side effects. For example, an opioid agonist-β-adrenergic-antagonist compound might enhance opioid activity (since adrenergic antagonists appear to function as opioid enhancers as well as do agonists) yet spare the patient β-adrenergic activation that might affect the heart. Alternatively, such a tethered drug might be exactly what is needed for treating a patient in need of a β blocker as well as analgesia. Conversely, since β-adrenergic agonists such as albuterol or salbutamol are sometimes used in combination with opioids such as naloxone or naltrexone, a tethered compound might provide enhanced COPD relief [153], asthma relief for heroin-induced bronchospasm [154] or for asthmatics who are becoming resistant to rescue inhalers without activating opioid mechanisms, a phenomenon already demonstrated with adrenergic-ascorbate combinations [84,88,155,156] and linked compounds [114]. The point of Table 2 is that novel tethered compounds will permit many new ways of manipulating the phenomenon of adrenergic-opioid co-enhancement and co-antagonism through their effects both on the individual receptor subtypes as well as by manipulating heterodimerization states. 

## Figures and Tables

**Figure 1 pharmaceuticals-15-00214-f001:**
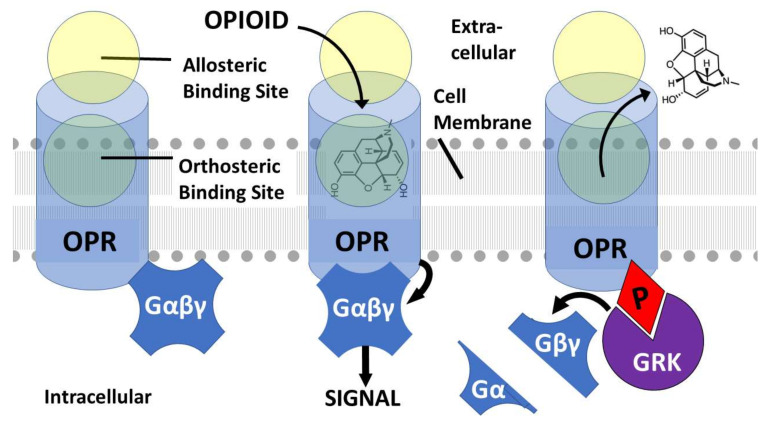
A simplified description of opioid receptor (OPR) activation and desensitization in the presence of an opioid agonist such as morphine (illustrated) or any other OPR ligand. OPR are activated by binding an appropriate ligand into their highly specific orthosteric binding site (**left**). OPR also contain extracellular allosteric binding sites (**left**). Orthosteric ligand binding (**center**) results in a conformational change in the receptor that allows an intracellular guanine nucleotide-binding protein (G protein: Gαβγ) to exchange a guanine diphosphate (GDP) for a guanine triphosphate (GTP), the GTP then permitting one of the G protein subunits to dissociate (**right**) and act as an intracellular second messenger to activate downstream functions. The dissociation of the G protein also activates the phosphorylation (red diamond P) of the remaining subunits by G protein-coupled receptor kinases (GRKs), which results in the internalization of the receptor, and thus the downregulation of receptor activity. Repeated or continuous stimulation of GPCR results in tolerance to the ligand as a result of this downregulation so that increasing concentrations of drug are needed to produce the previous effect.

**Figure 2 pharmaceuticals-15-00214-f002:**
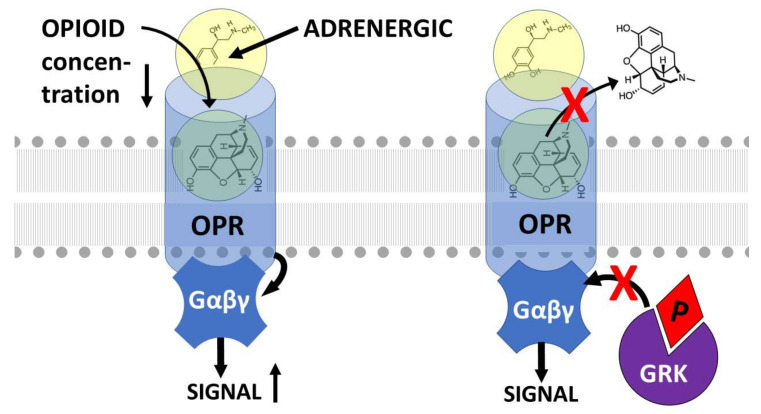
Illustration of the effects of binding an adrenergic drug (illustrated here by epinephrine, but not limited to it) to the allosteric site of the opioid receptor (OPR) in the presence of an opioid drug that binds to the orthosteric site (refer to Figure 1). (**left**) Binding of an opioid into the orthosteric site results, as in Figure 1, in conformational changes in the OPR that result in G protein (Gαβγ) activation of intracellular signaling. However, in the presence of allosteric binding of an adrenergic drug, such signaling is activated at significantly lower concentrations of opioid while the degree of signal activation is increased for any given concentration of opioid. (**right**) A second effect of allosteric binding of an adrenergic drug in the presence of orthosteric binding of an opioid is to prevent release of the opioid from the allosteric site (top red x) and to retain the OPR in its high activity, high affinity conformation. This conformation resists (lower red x) G-protein related kinase (GRK) phosphorylation (P) of the OPR (bottom red arrow) and therefore its downregulation. Thus, adrenergic binding to the extracellular allosteric site on OPR results in increased receptor activation at lower opioid concentrations and increased duration of activation. Model based on data from [85,86].

**Figure 3 pharmaceuticals-15-00214-f003:**
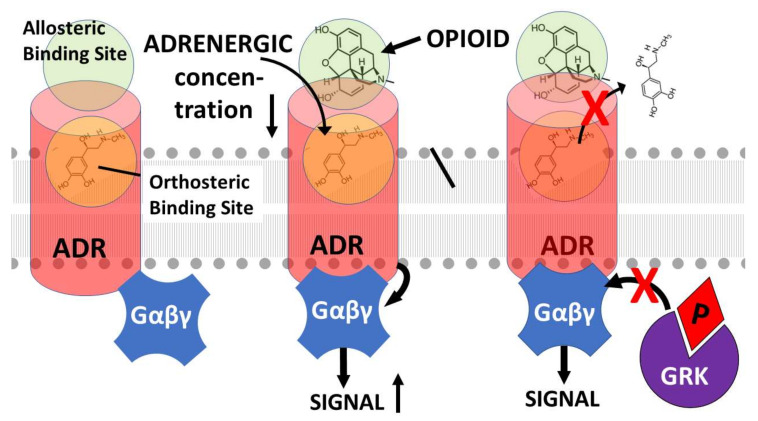
Model of the effects of opioid binding to the extracellular allosteric site of the adrenergic receptor (ADR). ADR, like OPR (Figure 1 and Figure 2), are G-protein coupled receptors (GPCR) and are activated and downregulated in the same manner. Thus, binding of an adrenergic drug (epinephrine is illustrated but represents any adrenergic compound) into the orthosteric binding site of the ADR results in a conformational change in the receptor that activated G-protein (Gαβγ) mediated intracellular signaling. Opioid binding into the ADR extracellular allosteric site decreases the concentration of adrenergic drug required to activate the ADR while increasing the signaling achieved at any given concentration of adrenergic drug. Binding of the opioid into the allosteric site also prevents (red x) release of the adrenergic drug from the orthosteric site as well as G-protein related kinase (GRK) phosphorylation (P) of the ADR and thus its downregulation. Thus, as in Figure 1 and Figure 2, the result of co-stimulation of ADR with an adrenergic and opioid drug is to enhance the adrenergic activity (whether agonistic or antagonistic) at lower concentrations of the adrenergic drug and to increase the duration of the resulting activity. Model based on data from [88,89,114].

**Figure 4 pharmaceuticals-15-00214-f004:**
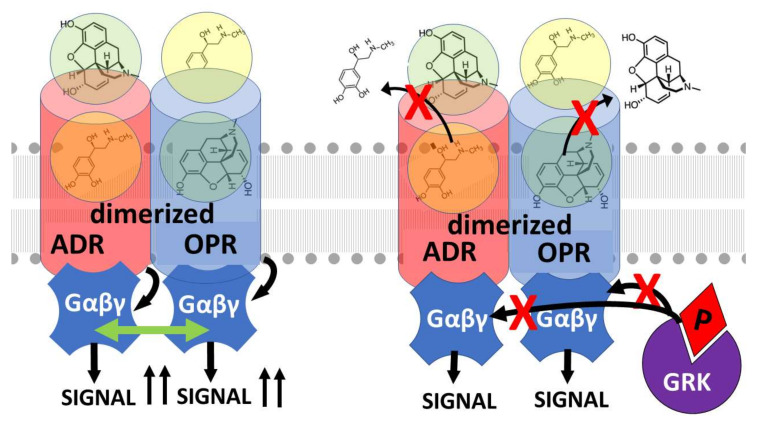
Model of the effects of joint stimulation of opioid receptors (OPR) and adrenergic receptors (ADR) on receptor heterodimerization. In addition to the allosteric enhancement of OPR and ADR in the presence of both of their ligands, this enhancement also potentiates OPR-ADR dimerization and oligomerization. (**left**) This dimerization brings the pairs or receptors into contact with each other further increasing the degree of receptor activation at any given drug concentration as well as the duration of activation. Cooperation between the G-proteins (Gαβγ) (green arrow) may also occur. (**right**) The dimerization further enhances the resistance of the receptors to downregulation by preventing release of the ligands from their orthosteric sites (top red x’s) and phosphorylation (P) of the G proteins by G-protein-related kinases (GRK) (bottom red x’s). Thus, stimulation of OPR and ADR simultaneously results in both allosteric- and dimerization-mediated enhancement of opioid activity, decreasing the amount of opioid required to activate analgesia as well as increasing the duration of opioid activity.

**Figure 5 pharmaceuticals-15-00214-f005:**
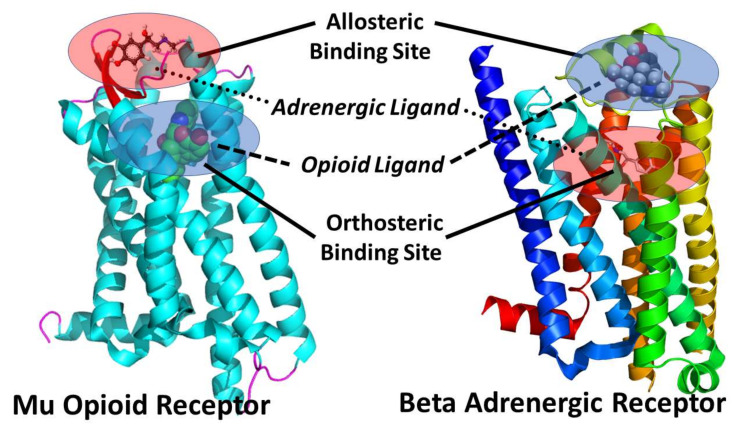
Ribbon models of the mu opioid receptor (MOR) and the β adrenergic receptor (Wikimedia Commons) illustrating the locations of the orthosteric and allosteric binding sites on each based on the references provided in this paper in Section 2. The proximity of the allosteric site to the orthosteric site in both types of receptors means that bitopic opioid–adrenergic drugs will have relatively short tethers. The degree to which a tethered heterobitopic compound will orient the adrenergic and opioid components relative to each other so that they are specific for either opioid receptors or adrenergic receptors or both is as yet unknown and probably depends on the attachment site of the linker to each component as well as the flexibility of the linker.

**Figure 6 pharmaceuticals-15-00214-f006:**
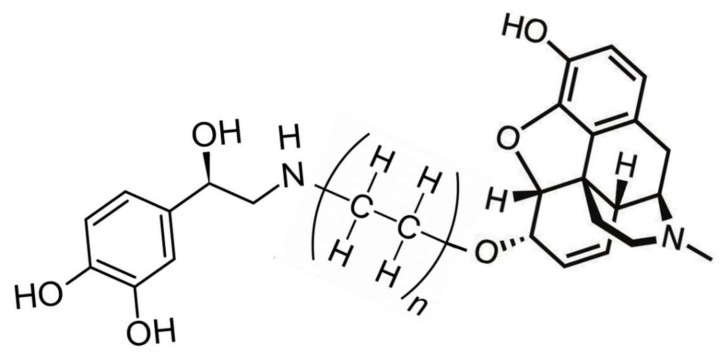
An example of a bitopic adrenergic-opioid tethered drug. The adrenergic module (**left**) is epinephrine. The opioid module (**right**) is morphine. The tether (**center**) is polyethylene. Given experimental and clinical studies of the effects of treating animals and patients with epinephrine and morphine together, it can be predicted that such a drug would have enhanced opioid activity of much greater duration at significantly lower doses than morphine alone and therefore produce better analgesia with less side effects. Such a drug might be particularly useful for local analgesia applications such an epidural or topical applications. Any other adrenergic agonist could potentially replace epinephrine, any opioid agonist could potentially replace morphine, and tethers are not limited to polyethylene (see text).

**Figure 7 pharmaceuticals-15-00214-f007:**
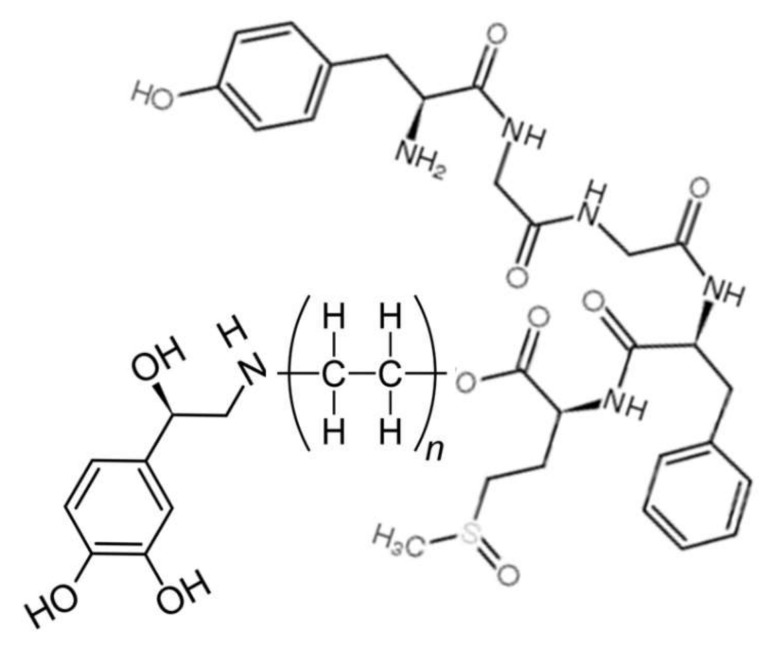
An example of a bitopic adrenergic-opioid tethered drug. The adrenergic module (**left**) is epinephrine. The opioid module (**right**) is methionine enkephalin. The tether (**center**) is polyethylene. This structure illustrates the fact that endogenous opioid such as the enkaphalins and their modified versions (e.g., D-Ala2, N-MePhe4, Gly-ol]-enkephalin [DAMGO]) may also provide the basis for novel, biased, and enhanced tethered drugs.

**Figure 8 pharmaceuticals-15-00214-f008:**
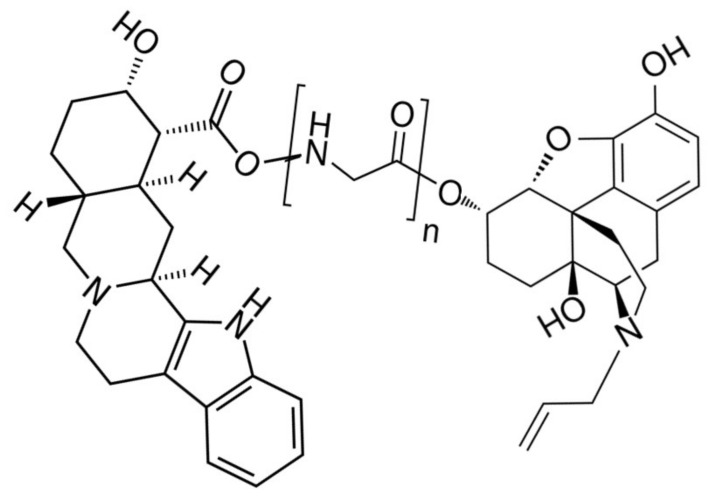
Example of a biotopic opioid–adrenergic drug with dual antagonistic activity. The adrenergic module is yohimbine (**right**), an α-adrenergic antagonist. The opioid module (**left**) is α naloxol, a potent opioid receptor antagonist. The tether (**center**) is poly-glycine. Such a dual-antagonist drug may produce significantly better opioid antagonism than naloxone or naltrexone without activating adrenergic receptors and thus be particularly useful for treating opioid overdoses or side effects. Other adrenergic antagonists (e.g., propranolol) and opioid antagonists (e.g., naltrexone), and additional linkers, offer additional design options.

**Figure 9 pharmaceuticals-15-00214-f009:**
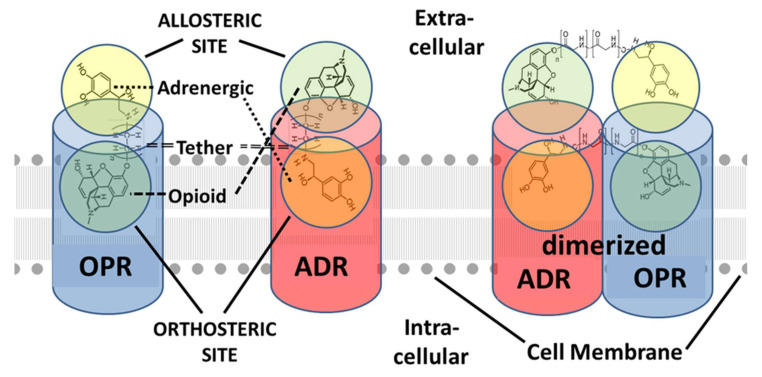
Models of how tethered biotopic opioid–adrenergic drugs may interact with opioid receptors (OPR) and adrenergic receptors (ADR). (**left**) Biotopic drugs with short tethers will be able to bind to both the OPR and the ADR separately (see Figure 5 for further discussion). At the OPR, the opioid module will bind to the orthosteric site and the adrenergic module to the allosteric site, resulting in enhancement of opioid agonist or antagonist activity. At the ADR, the opioid module will bind to the allosteric site and the adrenergic module to the orthosteric site resulting in the enhancement of adrenergic agonistic or antagonistic activity. It should therefore be possible to design tethered bitopic drugs that selectively activate both the OPR and ADR simultaneously, the OPR rather than the ADR, or the ADR rather than the OPR, depending on whether the opioid and adrenergic modules are agonists or antagonists (see Table 2). The length of the tether and the way it orients the opioid and adrenergic modules in relation to each other may also help to determine receptor biasing. (**right**) Simultaneous activation of OPR and ADR naturally leads to their heterodimerization. However, bitopic drugs with long tethers will be able to potentiate and stabilize heterodimerization of OPR with ADR, enhancing opioid and/or adrenergic agonistic or antagonistic activity. While design of tethers for optimizing heterodimerization will naturally focus on linkers that place both the opioid and adrenergic modules in their respective orthosteric binding sites, it is likely that further enhancement of receptor activity will result from binding of the tethered drug to the extracellular allosteric sites as well. (NOT SHOWN) It may also be possible to combine the two approaches to biotopic drug design shown here to link the type of short-tethered bitopic drugs shown on the left with long linkers such as those illustrated on the right to produce drugs that optimize binding to receptor orthosteric and allosteric sites simultaneously with optimizing heterodimerization of receptors.

**Figure 10 pharmaceuticals-15-00214-f010:**
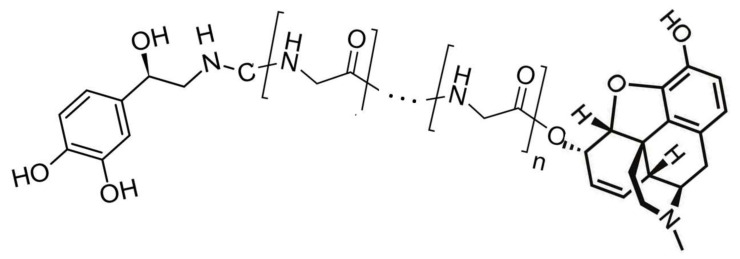
Model of a bitopic, tethered adrenergic-opioid drug optimized for enhancing opioid receptor (OPR) heterodimerization with adrenergic receptors (ADR). The adrenergic module illustrated (**left**) is epinephrine. The opioid module illustrated (**right**) is morphine. The tether (**center**) is poly-glycine. Such a drug would optimize MOR dimerization with β-adrenergic receptors. Other choices of adrenergic modules could optimize different ADR subtypes, while other choices of opioids could optimize delta or kappa opioid receptor dimerization and thus produce enhanced, biased activity of choice.

**Figure 11 pharmaceuticals-15-00214-f011:**
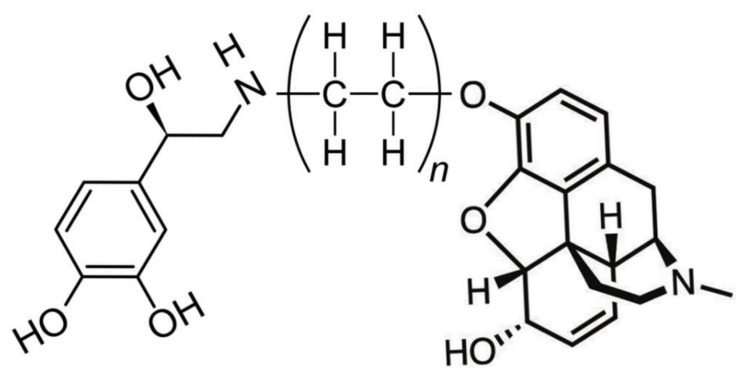
An example of a heterobitopic adrenergic agonist-inactive opioid tethered drug. The adrenergic module (**left**) is epinephrine. The opioid module (**right**) is morphine. The tether (**center**) is polyethylene. Tethering the morphine to the phenol group should inactivate the morphine at the orthosteric site, blocking the receptor without activating it. Such a drug might be useful for improving asthma treatments, for example, by increasing specific binding and activation of b adrenergic receptors in trachealis tissue without inducing analgesia or other opioid effects. Alternatively, a known antagonist such as naloxone or naltrexone could be tethered to the adrenergic agonist. Conversely, biotopic drugs with active opioid agonists tethered to adrenergic antagonists (or tethered to the adrenergic moiety in such a way as to inactivate it) could provide additional options for increasing the affinity and specificity of bitopic drugs to one receptor type without activating the other.

**Table 1 pharmaceuticals-15-00214-t001:** Table summarizing the main clinical effects of combining various adrenergic agonists and antagonists (left-hand column) with opioid agonists (second column from the left). Notably, combinations of adrenergic drugs with opioids have mainly focused on mu-opioid receptor agonists (MOR), with very few studies of their effects on kappa-opioid receptor agonists (KOR) or delta-opioid receptor agonists (DOR). Additionally, it appears that the effects of adrenergic agonists and antagonists on opioid antagonists have not been explored in clinical studies. The numbers in the columns refer to the citations in the References.

Adrenergic Drug	Opioid Drug	Improved Anesthesia or Analgesia	Increases Anesthesia Duration	Opioid Sparing	Prevents Opioid Tolerance	Reverses Opioid Tolerance	No Benefits Reported
Epinephrine (α and β agonist)	Morphine (MOR)	[20,21,25,26,27,28,29]	[20,21,25,26,27,28,30,31]	[17,20,25,29,36,39,40]	[36,37,38]		
Epinephrine (α and β agonist)	Sufentanil (MOR)						[34]
Noradrenaline (α1 and 2 agonist)	Morphine (MOR)	[14,58]	[14,58]				
Clonidine (α2 agonist)	Morphine (MOR)	[14,20,21,27,30,32,33,58]	[14,20,21,22,27,30,32,33,58]	[17,36,39,40,58]	[36,38,42,43,58]	[42,43]	
Clonidine (α2 agonist)	Fentanyl (MOR)	[20,21,22,31,58]	[20,31,58]				
Clonidine (α2 agonist)	Meperidine (MOR, KOR)	[58]	[58]				
Clonidine (α2 agonist)	DAMGO (MOR)	[58]	[58]				
Clonidine (α2 agonist)	Sufentanil (MOR)						[34]
Guanfacine (α2A agonist)	Morphine (MOR)				[35,36,37,38]	[35]	[61]
Guanfacine (α2A agonist)	DAMGO (MOR)						[[61]
Guanfacine (α2A agonist)	Deltorphin (DOR)	[61]					
Dexmedetomidine (α2 agonist)	Morphine (MOR)	[40,61]	[61]		[35,36,37,38,41,42,43]	[35,41,45]	
Prazosin (α1 antagonist)	Morphine (MOR)			[50]	[42,43,50]	[43]	
Phentolamine (α1 antagonist)	Morphine (MOR)			[52]	[52]		
BRL44408 (α2 antagonist)	Morphine (MOR)	[51]		[51]	[51]		
Idazoxan (α2 antagonist)	Morphine (MOR)	[39]		[39,58]	[58]		
Esmolol (β1 agonist)	Morphine (MOR)	[55,56]		[53,54,55,56,57]	[53,54,55,56,57]		
Propranolol (β1 and 2 antagonist)	Morphine (MOR)			[52]	[52]		

**Table 2 pharmaceuticals-15-00214-t002:** General overview of the various ways in which tethered, bitopic opioid–adrenergic drugs can be predicted to produce biased agonist or antagonist activity. OPR is the opioid receptor; ADR is the adrenergic receptor. Note that experimental and clinical data reviewed in the text demonstrate that in most cases the binding of adrenergic or opioid antagonists as well as agonists to the allosteric binding site of the complementary receptor results in enhancement of orthosteric-bound drug. Thus, in general, the orthosteric sites determine they type of activity (agonistic or antagonistic) while the allosteric sites mediate enhancement of that activity regardless of the orthosteric activity of the drug.

Opioid Ligand	Adrenergic	Effect on OPR	Effect on ADR	Effect on OPR-ADR Dimerization	Effect on OPR-OPR Dimerization	Effect on ADR-ADR Dimerization
	Ligand					
AGONIST	AGONIST	ENHANCED AGONIST	ENHANCED AGONIST	ENHANCED Dimerization	ENHANCED Dimerization	ENHANCED Dimerization
AGONIST	*Antagonist*	ENHANCED AGONIST	*ENHANCED Antagonist*	*Blocked Dimerization?*	ENHANCED Dimerization	*Blocked Dimerization*
*Antagonist*	AGONIST	*ENHANCED Antagonist*	ENHANCED AGONIST	*Blocked Dimerization?*	*Blocked Dimerization*	ENHANCED Dimerization
*Antagonist*	*Antagonist*	*ENHANCED Antagonist*	*ENHANCED Antagonist*	*Blocked Dimerization*	*Blocked Dimerization*	*Blocked Dimerization*

## Data Availability

Data sharing not applicable.

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
