# Peer review of "Biased, Bitopic, Opioid–Adrenergic Tethered Compounds May Improve Specificity, Lower Dosage and Enhance Agonist or Antagonist Function with Reduced Risk of Tolerance and Addiction"

_pharmaceuticals, 2022, doi:10.3390/ph15020214_

Round 1

Reviewer 1 Report

In his paper "Biased, Bitopic, Opioid-Adrenergic Tethered Compounds May Improve Specificity, Lower Dosage and Enhance Agonist or Antagonist Function with Reduced Risk of Tolerance and Addiction", Prof. Root-Bernstein reviews published evidence for allosteric interactions between adrenergic and opioid compounds. The Review aims to provide clues to the tethered molecules design, which could potentially augment opioid receptor activity to reduce beta-arrestin mediated run-down.

Line 50 - "For example..." phrase seems not connected with its surroundings. Prior to it, the author writes about adrenergic/opioid combinations. But then, suddenly, says about opioid combinations with anaesthetics with no references. Indeed, some adrenergic compounds show analgesic effects (e.g. cocaine), but the term "anaesthetics" is not limited by adrenergic compounds.

Figure 1 needs some adjustments: Gαβγ trimer dissociation is correctly described in the legend but not in the figure. It would be better to show G heterotrimer dissociation to α and βγ explicitly.

Section 2 would benefit from a table summarizing various adrenergic compounds actions on opioid receptors and vice versa. Readers would be happy to see a concise summary table with names of active and non-active compounds.

Overall, the Review gives a good perspective on the tethered opioid/adrenergic/adrenolytic ligands as potential drugs.

Author Response

REVIEWER 1

Journal

Pharmaceuticals (ISSN 1424-8247)

Manuscript ID

pharmaceuticals-1545352

Type

Review

Title

Biased, Bitopic, Opioid-Adrenergic Tethered Compounds May Improve Specificity, Lower Dosage and Enhance Agonist or Antagonist Function with Reduced Risk of Tolerance and Addiction

Authors

Robert Root-Bernstein *

Special Issue

Drug Candidates for Anesthesia and Analgesia

Author's Reply to the Review Report (Reviewer 1)

Please provide a point-by-point response to the reviewer’s comments and either enter it in the box below or upload it as a Word/PDF file. Please write down "Please see the attachment." in the box if you only upload an attachment. An example can be found here.

* Author's Notes to Reviewer

p

Word / PDF

or

Review Report Form

Open Review

(x) I would not like to sign my review report

( ) I would like to sign my review report

English language and style

( ) Extensive editing of English language and style required

( ) Moderate English changes required

(x) English language and style are fine/minor spell check required

( ) I don't feel qualified to judge about the English language and style

Is the work a significant contribution to the field?             

Is the work well organized and comprehensively described?        

Is the work scientifically sound and not misleading?         

Are there appropriate and adequate references to related and previous work?   

Is the English used correct and readable?             

Comments and Suggestions for Authors

In his paper "Biased, Bitopic, Opioid-Adrenergic Tethered Compounds May Improve Specificity, Lower Dosage and Enhance Agonist or Antagonist Function with Reduced Risk of Tolerance and Addiction", Prof. Root-Bernstein reviews published evidence for allosteric interactions between adrenergic and opioid compounds. The Review aims to provide clues to the tethered molecules design, which could potentially augment opioid receptor activity to reduce beta-arrestin mediated run-down.

Line 50 - "For example..." phrase seems not connected with its surroundings. Prior to it, the author writes about adrenergic/opioid combinations. But then, suddenly, says about opioid combinations with anaesthetics with no references. Indeed, some adrenergic compounds show analgesic effects (e.g. cocaine), but the term "anaesthetics" is not limited by adrenergic compounds.

Absolutely correct; offending line deleted.

Figure 1 needs some adjustments: Gαβγ trimer dissociation is correctly described in the legend but not in the figure. It would be better to show G heterotrimer dissociation to α and βγ explicitly.

Revised

Section 2 would benefit from a table summarizing various adrenergic compounds actions on opioid receptors and vice versa. Readers would be happy to see a concise summary table with names of active and non-active compounds.

Added

Overall, the Review gives a good perspective on the tethered opioid/adrenergic/adrenolytic ligands as potential drugs.

Submission Date

29 December 2021

Date of this review

10 Jan 2022 12:57:46

© 1996-2022 MDPI (Basel, Switzerland) unless otherwise sta

Reviewer 2 Report

This review described in detail the mechanisms underlying the enhancement of analgesic effect by combining adrenergic and opioid drugs. This combination may provide the basis for the development of new classes of tethered bitopic drugs with biased activity and fewer side effects. Therefore, I think that this review deals with a timely subject and useful for many readers. However significant revision is required before the manuscript is acceptable for publication.

The evidence for adrenergic binding sites in opioid receptors simply does not exist from either a structural and/or computational biology perspective. Evidence for opioid-adrenergic sites and/or one drug class allosterically modulating each other does exist. I am uncomfortable with Figures 1-5 presented in a way that suggests that allosteric sites at MOP are present which the α2 or β2 adrenergics may occupy and bind. I do not question that MOP and α2 do have synergy and modulate each other’s actions. Similarly, I am convinced that heteromization between opioids and α2 could be a mechanism by which these drugs modulate each other’s actions.

Given this, I propose stating that the author hypothesizes MOP/α2 actions to be a consequence of an unidentified allosteric site in the MOP for example. This however needs to be validated by cryoEM or X-ray structures or computationally by MD simulations.

MOP allosteric modulators are known and characterized by Traynor and Filizola groups. Although well characterized, where these ligands bind continues to be unknown. Some references to be included in the review are; PMID: 34153316, 23754417 and 33846240. An additional allosteric site was computationally, synthetically and pharmacologically validated in MOP and KOP using the KOR x-ray structure residing in TM5-ECL2. Cite PMID: 33555255 in this manuscript.

Other major criticisms are centered on the chemistry of compounds proposed under figures 6-10.

In figure 6: Move epinephrine to the 6α-OH position. As drawn with the phenol blocked, the compound will be inactive at MOP.

Figure 7: Simply delete this figure. I cannot thinking of ways of linking an opioid peptide to an adrenergic. Blocking the phenol of tyrosine will render the bivalent ligand inactive at MOP again. If required use the C-terminus end and not the N-terminus for design. Peptides are not good starting materials for drug discovery work.

Figure 8: Use 6α-naloxol or 6β-naloxol template and move yohimbine to the 6-position. As drawn with the phenol blocked, the compound will be inactive at MOP.

Figure 10: Move linker and epinephrine to the 6α-OH end. As drawn with the phenol blocked, the compound will be inactive at MOP.

 Minor Points:

  • Different abbreviations and symbols for opioid receptor names were used in the manuscript. Therefore, I highly recommend using their approved nomenclature (MOP, KOP, and DOP) for (mu, kappa, and delta) opioid receptors, respectively.
  • Different abbreviations and names were also used for beta adrenergic receptor. Sometimes author used beta and sometime the author used β. The consistency of the receptor names is required.
  • In Figures (1, 2, 3, and 4), since the chemical structure of morphine was used as an example for opioid agonist, opioid receptor (OPD) should be replaced by MOP to be more specific.
  • In section 3, I highly suggest adding a table to summarizes various bitopic compounds with different linker and their effects (therapeutic effects vs. side effects).
  • Pg8 line 321: Evidence for INTA being a bitopic is currently weak inspite of authors in these papers using that terminology. It maybe better to remove the INTA references or state the weakness in the hypothesis.
  • Pg 10 line 366: Use “bivalent” instead of bitopic for Portoghese’s synthesis of naloxone based ligands
  • Pg 13 line 455: 6’GNTI is currently belived to be a KOP partial agonist. See PMID: 22736766 and 23775075. It maybe better to remove the GNTI part.

Author Response

Journal

Pharmaceuticals (ISSN 1424-8247)

Manuscript ID

pharmaceuticals-1545352

Top of Form

Review Report Form

Open Review

English language and style

( ) Extensive editing of English language and style required
( ) Moderate English changes required
(x) English language and style are fine/minor spell check required
( ) I don't feel qualified to judge about the English language and style

Is the work a significant contribution to the field?

Is the work well organized and comprehensively described?

Is the work scientifically sound and not misleading?

Are there appropriate and adequate references to related and previous work?

Is the English used correct and readable?

Comments and Suggestions for Authors

This review described in detail the mechanisms underlying the enhancement of analgesic effect by combining adrenergic and opioid drugs. This combination may provide the basis for the development of new classes of tethered bitopic drugs with biased activity and fewer side effects. Therefore, I think that this review deals with a timely subject and useful for many readers. However significant revision is required before the manuscript is acceptable for publication.

The evidence for adrenergic binding sites in opioid receptors simply does not exist from either a structural and/or computational biology perspective. Evidence for opioid-adrenergic sites and/or one drug class allosterically modulating each other does exist. I am uncomfortable with Figures 1-5 presented in a way that suggests that allosteric sites at MOP are present which the α2 or β2 adrenergics may occupy and bind. I do not question that MOP and α2 do have synergy and modulate each other’s actions. Similarly, I am convinced that heteromization between opioids and α2 could be a mechanism by which these drugs modulate each other’s actions.

The comments by this Reviewer are technically and specifically correct but ignore the fact that there are a variety of studies demonstrating direct binding of adrenergics to opioid receptors as well as to fragments of the receptors that help to locate the putative binding sites. While one may prefer one type of evidence over another, it is not acceptable to ignore the evidence that already exists. Moreover, the Reviewer’s comments make no sense within the context of the paper as a whole and in light of the Reviewer’s own subsequent comments below.  Let me break this down. 

The Reviewer is correct that there is no direct evidence for adrenergic binding sites on opioid receptors from structural or computation biology studies. There is, however, direct evidence of binding of adrenergic drugs to opioid receptors from my and several other laboratories, utilizing a variety of techniques, and, in my case, we have specifically explored whether that binding is to extracellular loop regions of the receptor. The Reviewer is similarly correct that these studies do not PROVE that adrenergic drugs allosterically enhance MOP activity although the paper reviews other studies that make this highly likely. Thus, I am very willing to add language that points out the limitations of the evidence for allosteric, adrenergic binding sites on opioid receptors but I bridle at withdrawing the presence of those sites on FIGURES 1-5 for the simple reason that the majority of the drug designs that follow are predicated on the existence of such allosteric binding sites. Oddly, the Reviewer has not objected to the proposed compounds that have short tethers, which require the existence of an allosteric binding site for the adrenergic component very close to the orthosteric binding site for the opioid. Since the Reviewer does not object to the tethered drug designs, which ASSUME the existence of allosteric binding sites near the opioid binding site, I have to assume that the Reviewer is actually willing to consider the possibility of the existence of these allosteric sites essentially in the locations indicated (indeed, see more below).

I also point out that Traynor’s group has proposed an allosteric site on opioid receptors located in essentially the same place as that proposed in this (and our previous papers), which I now cite in appropriate places in my manuscript:

Livingston KE, Traynor JR. Allostery at opioid receptors: modulation with small molecule ligands. Br J Pharmacol. 2018 Jul;175(14):2846-2856. doi: 10.1111/bph.13823. Epub 2017 Jun 7. PMID: 28419415; PMCID: PMC6016636

See particularly Figure 5 of the above paper, which is nearly identical in conception to my figures. See also:

Livingston KE, Stanczyk MA, Burford NT, Alt A, Canals M, Traynor JR. Pharmacologic Evidence for a Putative Conserved Allosteric Site on Opioid Receptors. Mol Pharmacol. 2018 Feb;93(2):157-167. doi: 10.1124/mol.117.109561. Epub 2017 Dec 12. PMID: 29233847; PMCID: PMC5767684.

Additionally, the following group has also proposed, through evidence from in silico and experimental studies (again the type suggested by the Reviewer), the existence of an allosteric binding site on the extracellular loops of opioid receptors:

Bartuzi D, KÄ™dzierska E, Kaczor AA, Schmidhammer H, Matosiuk D. Novel Positive Allosteric Modulators of µ Opioid Receptor-Insight from In Silico and In Vivo Studies. Int J Mol Sci. 2020 Nov 11;21(22):8463. doi: 10.3390/ijms21228463. PMID: 33187107; PMCID: PMC7697543.

Bartuzi D, Kaczor AA, Matosiuk D. Activation and Allosteric Modulation of Human μ Opioid Receptor in Molecular Dynamics. J Chem Inf Model. 2015 Nov 23;55(11):2421-34. doi: 10.1021/acs.jcim.5b00280. Epub 2015 Nov 11. PMID: 26517559.

In sum, there is nothing particularly novel about FIGURES 1-5 that would make it necessary to delete or modify them. Instead, I am adding language that points out the prior work of the Traynor, Filizola and Matosiuk groups as further evidence for the likelihood of an allosteric binding site on the extracellular side of the orthosteric site and near or contiguous with portions of the extracellular loops.

Given this, I propose stating that the author hypothesizes MOP/α2 actions to be a consequence of an unidentified allosteric site in the MOP for example. This however needs to be validated by cryoEM or X-ray structures or computationally by MD simulations.

I have no problem adding language to the effect that the location of the proposed allosteric site, and the details of its specificity, will require further structural and computational studies.  Given my own group’s studies and those of Traynor and Filizola (see next Reviewer comment below), I am not willing to go so far as to state that the existence of allosteric binding to opioid receptors is “hypothetical” or is “unidentified”.  Multiple groups have proposed the same general site using multiple approaches.

MOP allosteric modulators are known and characterized by Traynor and Filizola groups. Although well characterized, where these ligands bind continues to be unknown.

This statement is not entirely accurate. “Unknown”  implies that there is no evidence to help define where the binding site is likely to be. On the contrary, the studies mentioned are all consistent with each other and suggest the same basic location for the allosteric site (on the extracellular side of the orthosteric site, usually involving transmembrane region 5 and extracellular loop 2 at a minimum). Using the type of molecular dynamics methods the Reviewer him/herself advises, Filizola’s group has provided models of the proposed binding site for several allosteric modulators. See:

Shang Y, Yeatman HR, Provasi D, Alt A, Christopoulos A, Canals M, Filizola M. Proposed Mode of Binding and Action of Positive Allosteric Modulators at Opioid Receptors. ACS Chem Biol. 2016 May 20;11(5):1220-9. doi: 10.1021/acschembio.5b00712. Epub 2016 Feb 17. PMID: 26841170; PMCID: PMC4950826.

More importantly, other groups, including those cited above (Traynor’s and Matosiuk’s) have also used molecular dynamic methods to identify a very similar, if not identical, allosteric site. The confluence of binding studies and molecular dynamics studies therefore strongly suggests the probable location of the allosteric site.

Obviously, these models do not PROVE the location of the allosteric site but in combination with direct binding studies (many groups) and studies of ligand binding to receptor-derived peptides (our group), the existence and probable location of the allosteric ligand binding sites can at least be proposed with some confidence. Figured 1-5 stand.

Indeed, look at what the Reviewer says next, which would seem to undermine the very criticism that s/he has been making of Figures 1-5, but which actually confirm the existence of the allosteric site indicated in these Figures:

Some references to be included in the review are; PMID: 34153316, 23754417 and 33846240. An additional allosteric site was computationally, synthetically and pharmacologically validated in MOP and KOP using the KOR x-ray structure residing in TM5-ECL2. Cite PMID: 33555255 in this manuscript. [!!!!!] (My added underlining and exclamation points)

The references (below) requested by the Reviewer have been added to the manuscript in addition to the Traynor, Filizola and Matosiuk references above.

Burford NT, Clark MJ, Wehrman TS, Gerritz SW, Banks M, O'Connell J, Traynor JR, Alt A. Discovery of positive allosteric modulators and silent allosteric modulators of the μ-opioid receptor. Proc Natl Acad Sci U S A. 2013 Jun 25;110(26):10830-5. doi: 10.1073/pnas.1300393110. Epub 2013 Jun 10. PMID: 23754417; PMCID: PMC3696790.

Bisignano P, Burford NT, Shang Y, Marlow B, Livingston KE, Fenton AM, Rockwell K, Budenholzer L, Traynor JR, Gerritz SW, Alt A, Filizola M. Ligand-Based Discovery of a New Scaffold for Allosteric Modulation of the μ-Opioid Receptor. J Chem Inf Model. 2015 Sep 28;55(9):1836-43. doi: 10.1021/acs.jcim.5b00388. Epub 2015 Sep 15. PMID: 26347990; PMCID: PMC4703110.

Livingston KE, Traynor JR. Allostery at opioid receptors: modulation with small molecule ligands. Br J Pharmacol. 2018 Jul;175(14):2846-2856. doi: 10.1111/bph.13823. Epub 2017 Jun 7. PMID: 28419415; PMCID: PMC6016636.

Kandasamy R, Hillhouse TM, Livingston KE, Kochan KE, Meurice C, Eans SO, Li MH, White AD, Roques BP, McLaughlin JP, Ingram SL, Burford NT, Alt A, Traynor JR. Positive allosteric modulation of the mu-opioid receptor produces analgesia with reduced side effects. Proc Natl Acad Sci U S A. 2021 Apr 20;118(16):e2000017118. doi: 10.1073/pnas.2000017118. PMID: 33846240; PMCID: PMC8072371.

Uprety R, Che T, Zaidi SA, Grinnell SG, Varga BR, Faouzi A, Slocum ST, Allaoa A, Varadi A, Nelson M, Bernhard SM, Kulko E, Le Rouzic V, Eans SO, Simons CA, Hunkele A, Subrath J, Pan YX, Javitch JA, McLaughlin JP, Roth BL, Pasternak GW, Katritch V, Majumdar S. Controlling opioid receptor functional selectivity by targeting distinct subpockets of the orthosteric site. Elife. 2021 Feb 8;10:e56519. doi: 10.7554/eLife.56519. PMID: 33555255; PMCID: PMC7909954.

Other major criticisms are centered on the chemistry of compounds proposed under figures 6-10.

In figure 6: Move epinephrine to the 6α-OH position. As drawn with the phenol blocked, the compound will be inactive at MOP.

Very likely; figure therefore changed. (Note,  however, that this blockage provides a possibly novel way to produce inactive/antagonist opioids linked to active adrenergics, which has been added as a new FIGURE 11 to illustrate combination agonist/antagonist compounds.

Figure 7: Simply delete this figure. I cannot think of ways of linking an opioid peptide to an adrenergic. Blocking the phenol of tyrosine will render the bivalent ligand inactive at MOP again. If required use the C-terminus end and not the N-terminus for design.

The C-terminus idea is a good idea I’d considered in another iteration, so I’ve implemented it in the replacement figure here.

 Peptides are not good starting materials for drug discovery work.

Perhaps not, yet Traynor’s group (among many others) has performed many of their experiments on peptide opioids and allosteric peptide leads and one could certainly start with DAMGO instead, which is a standard for experimental studies (a suggestion now made explicit in the text). If nothing else, such drugs may be very useful for teasing out the mode of action (in vitro or in vivo) of the kinds of tethered compounds proposed here. References added to support using peptides as drug development candidates:

Olson KM, Traynor JR, Alt A. Allosteric Modulator Leads Hiding in Plain Site: Developing Peptide and Peptidomimetics as GPCR Allosteric Modulators. Front Chem. 2021 Oct 7;9:671483. doi: 10.3389/fchem.2021.671483. PMID: 34692635; PMCID: PMC8529114.

Kandasamy R, Hillhouse TM, Livingston KE, Kochan KE, Meurice C, Eans SO, Li MH, White AD, Roques BP, McLaughlin JP, Ingram SL, Burford NT, Alt A, Traynor JR. Positive allosteric modulation of the mu-opioid receptor produces analgesia with reduced side effects. Proc Natl Acad Sci U S A. 2021 Apr 20;118(16):e2000017118. doi: 10.1073/pnas.2000017118. PMID: 33846240; PMCID: PMC8072371.

A few lines addressing this problem have been added to the text and the references above incorporated into the reference list.

Figure 8: Use 6α-naloxol or 6β-naloxol template and move yohimbine to the 6-position. As drawn with the phenol blocked, the compound will be inactive at MOP.

Done

Figure 10: Move linker and epinephrine to the 6α-OH end. As drawn with the phenol blocked, the compound will be inactive at MOP.

Done

 Minor Points:

  • Different abbreviations and symbols for opioid receptor names were used in the manuscript. Therefore, I highly recommend using their approved nomenclature (MOP, KOP, and DOP) for (mu, kappa, and delta) opioid receptors, respectively.

Done, except that I prefer the standard MOR, KOR and DOR . Technically, MOP, KOP and DOP refer specifically to peptide receptors, whereas MOR, KOR and DOR are more generic in terms of referring to the fact the receptors can bind ligands other than peptide opioids.

  • Different abbreviations and names were also used for beta adrenergic receptor. Sometimes author used beta and sometime the author used β. The consistency of the receptor names is required.

Done. Same for alpha…

  • In Figures (1, 2, 3, and 4), since the chemical structure of morphine was used as an example for opioid agonist, opioid receptor (OPD) should be replaced by MOP to be more specific.

No.  One has to use SOME specific molecule in a figure but that doesn’t mean that the cartoon is meant to be limited to that molecule. The language in the Figure captions makes clear that the morphine molecule is a stand-in for any opioid ligand. Indeed, the same objection could be raised to the use of epinephrine, so I have added language to the effect that its presence, too, is merely a stand-in for any adrenergic compound.

  • In section 3, I highly suggest adding a table to summarize various bitopic compounds with different linker and their effects (therapeutic effects vs. side effects).

I don’t see how to do this. Since no heterobiotopic adrenergic-opioid compounds have yet been made, I assume the Reviewer means such a Table to review other bitopic compounds. The relevant opioid-based ones are already mentioned in the text. A general TABLE would require a much expanded review and even limiting it to just tethered compounds involving any opioid or adrenergic compound would be very difficult because of the huge diversity of applications and differences in study methods. I would not know where to begin. Adding on top of that an attempt to pull out of this data the effects of different linkers seems especially difficult given the diversity of ligands. Moreover, references 134-142 are referenced in the text as sources for studies on linker chemistries and activity profiles, and those papers have already reviewed the use of, and various benefits and problems associated with various linkers. I don’t see the point in covering the same ground again here.

  • Pg8 line 321: Evidence for INTA being a bitopic is currently weak in spite of authors in these papers using that terminology. It maybe better to remove the INTA references or state the weakness in the hypothesis.

I am not in a position to become involved in what is clearly a controversial issue for the Reviewer. Since the Reviewer has not provided any references that indicate what the weaknesses are, I am leaving the discussion of this topic as it stands and leave it to readers to draw their own judgements.

  • Pg 10 line 366: Use “bivalent” instead of bitopic for Portoghese’s synthesis of naloxone based ligands

Corrected

  • Pg 13 line 455: 6’GNTI is currently believed to be a KOP partial agonist. See PMID: 22736766 and 23775075. It maybe better to remove the GNTI part.

Deleted

 Submission Date

29 December 2021

Date of this review

19 Jan 2022 18:25:27

Bottom of Form

© 1996-2022 MDPI (Basel, Switzerland) unless otherwise stated

Round 2

Reviewer 1 Report

After the review, the author has augmented the manuscript and improved it. I have no objections to publishing the article in the present form.

Reviewer 2 Report

The author has been responsive to my critiques. Review is much improved now and should be acceptable for publication

I think author and reviewer are speaking the same language but using different terminology.

Two small changes are proposed 

Figure 8: Use  a -CONH linkage instead of an oxime linking yohimbine.

Figure 10: Assuming author meant "CH2" instead of "C". Also put a "n" next to the bracket after glycine linker.